# Probing Modified Gravity Theories with Scalar Fields Using Black-Hole Images

**Georgios Antoniou** [1,2,*], **Alexandros Papageorgiou** [3] and **Panagiota Kanti** [4]

1    Nottingham Centre of Gravity, Nottingham NG7 2RD, UK
2    School of Mathematical Sciences, University of Nottingham, University Park, Nottingham NG7 2RD, UK
3    Center for Theoretical Physics of the Universe, IBS, Daejeon 34126, Republic of Korea
4    Division of Theoretical Physics, Physics Department, University of Ioannina, GR 45110 Ioannina, Greece
*    Correspondence: georgios.antoniou@nottingham.ac.uk

**Abstract:** We study a number of well-motivated theories of modified gravity with the common overarching theme that they predict the existence of compact objects, such as black holes and wormholes endowed with scalar hair. We compute the shadow radius of the resulting compact objects and demonstrate that black hole images, such as that of M87* or the more recent SgrA* by the Event Horizon Telescope (EHT) collaboration, could provide a powerful way to constrain deviations of the metric functions from what is expected from general relativity (GR) solutions. We focus our attention on Einstein-scalar-Gauss–Bonnet (EsGB) theory with three well-motivated couplings, including the dilatonic and $Z_2$ symmetric cases. We then analyze the shadow radius of black holes in the context of the spontaneous scalarization scenario within EsGB theory with an additional coupling to the Ricci scalar (EsRGB). Finally, we turn our attention to spontaneous scalarization in the Einstein–Maxwell-Scalar (EMS) theory and demonstrate the impact of the parameters on the black hole shadow. Our results show that black hole imaging is an important tool for constraining black holes with scalar hair, and, for some part of the parameter space, black hole solutions with scalar hair may be marginally favored compared to solutions of GR.

**Keywords:** modified gravity; Gauss-Bonnet gravity; Einstein-Maxwell-scalar theory; scalarization; hairy black holes; black-hole shadow; wormhole shadow; black-hole imaging; Event Horizon Telescope

## 1. Introduction

Black holes, once considered a mere mathematical curiosity of Einstein's General Theory of Relativity, are now known to populate our Universe in vast numbers. Currently, they are met at two different scales: stellar black holes with masses in the approximate range of (5–70) $M_\odot$ and supermassive black holes residing at the center of galaxies with masses as large as $10^{10} M_\odot$. A black hole is the most lucid manifestation of how gravity behaves at the strong regime and can thus serve as a testbed for probing the fundamental theory of gravitational interactions.

Although General Relativity (GR) is a beautiful mathematical theory that has so far passed all experimental tests (see, for instance, [1–5] and reviews [6–9]), it is clear that it cannot provide all the answers to several persisting, open questions in gravity and cosmology: the existence of singularities, the unknown nature of dark matter and dark energy, the difficulty in quantizing gravity and unifying it with the remaining forces in nature, to mention a few. The common consensus among scientists is that GR is only a low-energy limit of a more fundamental theory of gravity [10,11]. As the structure of the final Quantum Theory of Gravity is still alluding us, the most usual approach taken in the meantime is that of the effective field theory: GR, a linear theory in terms of curvature, is now supplemented by higher gravitational terms, the presence of extra fields—mainly

scalar and gauge fields—and new couplings, including higher-derivative ones between matter and gravity.

Extending GR in this way unavoidably leads to a much richer range of gravitational solutions. To start with, new black-hole solutions in the context of modified theories of gravity have long been known to exist [12–20] (see also [21,22] for a review), by evading the no-hair theorems of GR [23–35] with a plethora of additional solutions have emerged during the last few years [36–58]. In addition, these modified theories also predict compact solutions other than black holes, such as traversable wormholes [59–80] and particle-like solutions [49,81–87]. The exciting prospect of having our Universe also populated by these compact objects perhaps does not seem so unlikely nowadays.

Our first task, however, is to probe the validity of the modified gravitational theories predicting all these new gravitational solutions. The properties of the observed black holes or the observable signals from processes associated with black holes can serve as a valuable tool for this purpose. Indeed, in the last few years, we have witnessed the detection of gravitational waves from the merging processes of stellar black holes [88–90] but also the imaging observations of the supermassive black holes residing at the center of the M87 galaxy [91–98] and of our own Galaxy [99–104]. These observations have been used extensively in the literature to probe the validity of General Relativity and to set limits and constraints on modified gravitational theories (see, for example, [105–128]). Capturing the horizon-scale image of Sagittarius A$^*$ in particular, the supermassive black hole located in the center of our own Galaxy, presents a number of advantages. First, due to its proximity, the mass-to-distance ratio of Sagittarius A$^*$ is much more accurately determined than that of M87$^*$. In addition, Sagittarius A$^*$ has a much smaller mass than M87$^*$; this allows us to test a curvature scale that lies between the low curvature scale of the massive M87$^*$ black hole and the high curvature scale of stellar black holes.

The main feature in the horizon-scale images of supermassive black holes is the bright photon ring that marks the boundary of a dark interior region, called the black-hole shadow [129]. The bright ring is formed by photon trajectories originating from parts of the universe behind the black hole that is gravitationally lensed by its gravitational field and directed toward our line of sight. These photons have impact parameters slightly larger than the ones that lead to their capturing in bound, circular orbits around the black hole. The quantitative characteristics of the shadow can be calculated in the context of either GR or a modified theory of gravity and compared to the observed value, thus probing the validity of the theory in question.

In this work, we consider a set of modified gravitational theories, with their common characteristic being the presence of a scalar field. This scalar field will be sourced by either gravitational terms, leading to induced or spontaneous scalarization, or gauge fields, leading to charged scalarized solutions. The presence of the scalar field modifies the gravitational background as well as the geodesic structure of the spacetime, including the photon trajectories and the size and shape of the black-hole shadow. We will initiate our analysis by deriving the connection between the metric components of the line element around a compact object and the theoretically expected shadow radius in a model-independent way. We will then apply this formalism in order to derive the shadow radius for compact objects in the Einstein-scalar-Gauss–Bonnet (EsGB) theory with three different forms of coupling function between the scalar field and the GB term in a variant of the EsGB theory with an additional coupling between the scalar field and the Ricci tensor, and finally, in the Einstein–Maxwell-Scalar (EMS) theory with three different forms of the coupling function between the scalar and the Maxwell fields again.

The validity of such scalar-tensor and tensor-scalar-vector theories could be probed by present and future observational bounds on shadow radii. To this end, after deriving the theoretically predicted shadow radii in each theory, we will employ, as an indicative example, the bounds on the deviation of the observed black-hole shadow of Sagittarius A$^*$ from that of the Schwarzschild solution [1], as these were derived by the Event Horizon Telescope [104] in a mass-scale independent form. We demonstrate that the black-hole

shadow bounds from Sagittarius A* can indeed impose restrictions on the parameter space or on the form of the coupling function of the scalar field in the aforementioned modified theories. However, the physical conclusions drawn depend very strongly on the particular EHT bound, or combination of EHT bounds, employed for this purpose. Thus, the use of individual bounds always allows amble parameter space where the majority of the modified theories considered are viable—in certain cases, they are even favored compared to General Relativity. In contrast, demanding that all EHT bounds are simultaneously satisfied significantly reduces the parameter space and, at times, eliminates it.

The outline of the paper is as follows. In Section 2, we present the formalism that allows us to derive the theoretically expected shadow radius in a model-independent way. We focus here on black holes and wormholes and demonstrate the differences in the shadows in each case. In Section 3, we provide a comprehensive review of the derivation of the existing observational bounds that quantify the deviation of the black-hole shadow from the expected GR result. Next, in Section 4, we apply the formalism of Section 2 to compute the shadow radius of black holes and wormholes derived in the context of the EsGB theory with three distinct coupling functions and demonstrate the effect of the bounds obtained by the EHT observations. We perform the same tasks in Section 5, where we turn our attention to some of the most well-established models of spontaneous scalarization, and in Section 6, we likewise analyze the EMS theory. We outline our conclusions in Section 7.

## 2. Shadow Radius of Compact Objects

In this section, we present the analytic formalism, which yields the expressions for the shadow radius of compact objects. We focus our analysis on solutions with spherical symmetry due to the significantly larger number of such scalarized solutions in the literature. In addition, as we will argue at the end of Section 3, the black-hole spin affects the observational value of the shadow rather feebly. In what follows, we first examine the case where the compact object is a black hole, and then we consider the scenario where the compact object is a wormhole.

### 2.1. Black Holes

We start by investigating the shadow size for a static and spherically symmetric configuration of the following form:

$$ds^2 = g_{tt}\,dt^2 + g_{rr}\,dr^2 + r^2 d\Omega^2\,. \tag{1}$$

We first need to locate the photon sphere for this background. To do that, we consider the trajectory of a photon. Since spherical symmetry is assumed, we can consider, without loss of generality, motion on the equatorial plane $\theta = \pi/2$. The Killing vectors associated with the symmetries of this spacetime are $\xi_1^\mu = (1,0,0,0)$ and $\xi_2^\mu = (0,0,0,1)$. Then, following [130], we can define the 4-momentum of a photon as $\tilde{k} = (k^t, k^r, k^\theta, k^\varphi)$, with $k^\theta = 0$ (from symmetry arguments). Then, the conserved quantities, i.e., the energy and angular momentum, are $E = -\xi_1^\mu k_\mu = -g_{tt}k^t$ and $L = \xi_2^\mu k_\mu = r^2 k^\varphi$, respectively. Moreover, the constraint $\tilde{k}^2 = 0$ fixes the $k^r$ component of the 4-momentum, so that we may finally write:

$$\tilde{k} = \left( -\frac{E}{g_{tt}}\,, \sqrt{-\frac{E^2}{g_{tt}\,g_{rr}} - \frac{L^2}{g_{rr}\,r^2}}\,, 0\,, \frac{L}{r^2} \right). \tag{2}$$

It is now straightforward to locate the radius for circular photon orbits by demanding $k^r = 0$ and $dk^r/dr = 0$. In terms of the impact parameter $b \equiv L/E$, these conditions yield

$$b^2 = -\frac{r^2}{g_{tt}}\bigg|_{r_{\rm ph}} = -\frac{r^3(g_{tt}\,g'_{rr} + g_{rr}\,g'_{rr})}{g_{tt}^2\,(r\,g'_{rr} + 2g_{rr})}\bigg|_{r_{\rm ph}}, \tag{3}$$

which can be simplified to give the equation for the photon circular orbit radius

$$\textit{Photon orbit radius:} \quad r_{\rm ph} = \frac{2g_{tt}}{g'_{tt}}\bigg|_{r_{\rm ph}}. \tag{4}$$

Our next step is to determine the shadow radius as observed by a far-away observer after lensing has been taken into account (see the left plot of Figure 1). For a null trajectory, we can write $g_{\mu\nu}\dot{x}^{\mu}\dot{x}^{\nu} = 0$, which, in turn, yields

$$g_{rr}\left(\frac{\dot{r}}{\dot{\varphi}}\right)^2 = -r^2 - g_{tt}\left(\frac{\dot{t}}{\dot{\varphi}}\right)^2, \tag{5}$$

where $E = -g_{tt}\dot{t}$ and $L = r^2\dot{\varphi}$. Therefore, we can equivalently solve for the radial deviation with respect to the polar angle

$$\left(\frac{dr}{d\varphi}\right)^2 = -\frac{r^2}{g_{rr}}\left(\frac{r^2}{g_{tt}\,b^2} + 1\right). \tag{6}$$

At the point of closest radial approach $r = r_0$, the equation above should vanish,

$$\frac{1}{b^2} = -\frac{g_{tt}}{r^2}\bigg|_{r_0}. \tag{7}$$

From Figure 1a, we can also easily deduce that

$$\cot\alpha = \frac{\sqrt{g_{rr}}}{r}\frac{dr}{d\varphi}\bigg|_{r_{\rm obs}} \xrightarrow{(6)} \sin^2\alpha = -\frac{g_{tt}\,b^2}{r^2}\bigg|_{r_{\rm obs}}. \tag{8}$$

Then, it is obvious that the angle for the shadow of the black hole is retrieved in the limit $r_0 \to r_{\rm ph}$. We assume that asymptotically far away, the spacetime is flat; therefore, $g_{tt} \to -1$. Then, for a far-away observer $\sin\alpha \approx \alpha$, so $\alpha_{\rm sh} = b_{\rm crit}/r_{\rm obs}$, where $b_{\rm crit}$ is the value of the impact parameter given in Equation (7) in the limit $r_0 \to r_{\rm ph}$. From Figure 1(a) and for $r_{\rm obs} \gg r_{\rm ph}$, we also have $\alpha_{\rm sh} \approx r_{\rm sh}/r_{\rm obs}$. Identifying the two expressions for $\alpha_{\rm sh}$, we can finally deduce that

$$r_{\rm sh} = b_{\rm crit} = \frac{r_{\rm ph}}{\sqrt{-g_{tt}(r_{\rm ph})}}. \tag{9}$$

For a Schwarzschild black hole, for example, where $g_{tt} = -(1 - 2M/r)$, Equation (4) readily gives $r_{\rm ph} = 3M$. Employing this result in Equation (9), we easily obtain that $r_{\rm sh} = 3\sqrt{3}M$.

### 2.2. Wormholes

By employing a different spherically symmetric metric, we can study other types of compact objects, which, in fact, exhibit different shadow properties. To this end, we thus consider the following alternative form of line element [71,72,131]

$$ds^2 = -e^{2v(l)}\,dt^2 + f(l)\,dl^2 + \left(l^2 + l_0^2\right)\left(d\theta^2 + \sin^2\theta\,d\varphi^2\right), \tag{10}$$

which describes a wormhole geometry with a throat located at $l_0$. In this spacetime, the conserved quantities are

$$E = -g_{tt}k^t = e^{2v}\frac{dt}{d\lambda}, \quad L = g_{\varphi\varphi}k^{\varphi} = \left(l^2 + l_0^2\right)\frac{d\varphi}{d\lambda}. \tag{11}$$

$$b^2 = \left(l_c^2 + l_0^2\right)e^{-2v_c}. \tag{12}$$

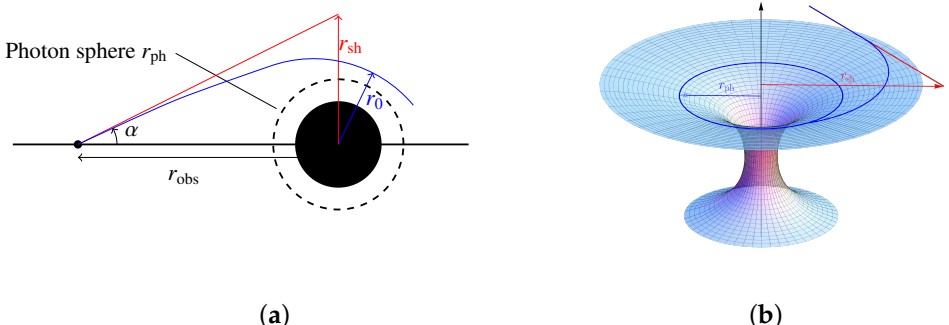

**Figure 1.** (**a**) Qualitative representation of a light ray reaching an observer at an angle $\alpha$, located at distance $r_{\rm obs}$ from the point singularity. The blue line traces a light ray escaping from a closed orbit around the black hole to infinity. The red line aligns with the inferred angle of approach for the light ray to an asymptotic observer. The point of closest approach for the light ray with respect to the black hole is located at $r = r_0$. If $r_0 = r_{\rm sh}$ the light ray escapes the photon sphere. The shaded, circular area denotes the interior of the black-hole horizon, while the dashed, circular line corresponds to the location of the photon sphere. (**b**) Same but for a wormhole geometry. Here we show the embedding diagram depicting a finite radius throat along the vertical axis. The blue line traces a light ray escaping from the photon sphere to infinity, while the red straight line corresponds to the inferred line of approach to an asymptotic observer.

In order to find the photon sphere(s), we demand, as in the black-hole case, that $k^l = 0$ and $dk^l/dl = 0$. These yield the following equation, which holds at the photon sphere(s):

$$v'(l_{\rm ph}) = \frac{l_{\rm ph}}{l_{\rm ph}^2 + l_0^2} \,. \tag{13}$$

Upon solving this, we obtain the radii for the circular photon orbits in this background, namely

$$l_{\rm ph} = \frac{1 \pm \sqrt{1 - 4\, l_0^2\, v_{\rm ph}'^2}}{2\, v_{\rm ph}'} \,. \tag{14}$$

Further, for a null trajectory, we now find

$$\left(\frac{dl}{d\varphi}\right)^2 = \frac{(l^2 + l_0^2)}{f}\left[-1 + \frac{(l^2 + l_0^2)}{e^{2v}\, b^2}\right] \,. \tag{15}$$

In order to reach the point of the closest approach $l = l_c$, where the above equation vanishes, the impact parameter must assume the following value

For the wormhole background (10), general Equation (8) for the lensing takes the form

$$\cot \alpha = \left.\sqrt{\frac{f(l)}{l^2 + l_0^2}}\,\frac{dl}{d\varphi}\right|_{l_{\rm obs}} \xrightarrow{(15)} \sin^2 \alpha = \left.\frac{e^{2v}\, b^2}{l^2 + l_0^2}\right|_{l_{\rm obs}} \,. \tag{16}$$

For $l_{\rm obs} \gg l_0, l_{\rm ph}$, asymptotic flatness demands that $v \to 0$. The wormhole shadow is retrieved again in the limit $l_c \to l_{\rm ph}$, for which $b \to b_{\rm crit}$ according to Equation (12). Thus, for a far-away observer, we obtain $a_{\rm sh} \approx b_{\rm crit}/l_{\rm obs}$. Moreover, from Figure 1b, we also find

$$a_{\rm sh} \approx \frac{r_{\rm sh}}{r_{\rm obs}} \approx \frac{\sqrt{l_{\rm sh}^2 + l_0^2}}{l_{\rm obs}} \,, \tag{17}$$

where we have used the fact that the space-like coordinate $l$ is related to the radial coordinate $r$ of the embedding diagram via the relation $l^2 = r^2 - l_0^2$. Thus, we can finally write

$$\sqrt{l_{\rm sh}^2 + l_0^2} = b_{\rm crit} = e^{-v(l_{\rm ph})}\sqrt{l_{\rm ph}^2 + l_0^2}. \tag{18}$$

One may apply the above formulae in the case of the Ellis–Bronnikov wormhole [59,60], where $e^{2v} = f = 1$. Then, Equation (14) gives $l_{\rm ph} = 0$, and thus there is only one circular photon orbit located around the throat. Then, in the limit $l_c \to l_{\rm ph}$, Equation (12) yields that $b_{\rm crit} = l_0$, and Equation (16) takes the simplified form

$$\sin^2 \alpha = \frac{l_0^2}{l_{\rm obs}^2 + l_0^2}, \tag{19}$$

which is exact and holds for all observers either far away or close by—this result is in agreement with Equation (72) of [132]. Applying the result $b_{\rm crit} = l_0$ in Equation (18), we obtain that $l_{\rm sh} = 0$, or equivalently that $r_{\rm sh} = l_0$. This behavior is expected to change for wormhole spacetimes with an $\ell$-dependent $g_{tt}$ metric component, as in Equation (10).

### 3. The EHT Bounds

The Event Horizon Telescope (EHT) is a Very-Long Baseline Interferometry (VLBI) array with Earth-scale coverage [91–98]. It is observing the sky at 1.3 mm wavelength and has, so far, managed to provide the horizon-scale image of the two supermassive black holes located at the center of the M87 galaxy and of our own galaxy. The diameter $\hat{d}_m$ of the bright photon ring surrounding the inner dark area—the most distinctive feature of these black-hole images—may be used to test theoretical predictions of both GR and modified theories. As noted above, in this work, we will be using the horizon-scale image of Sagittarius A$^*$. Following [104], one may write:

$$\hat{d}_m = \frac{\hat{d}_m}{d_{\rm sh}} d_{\rm sh} = \alpha_c \, d_{\rm sh} = \alpha_c \, (1 + \delta) \, d_{\rm sh,th} \,. \tag{20}$$

The diameter $\hat{d}_m$ is the value of the diameter of the photon ring obtained by using imaging and model fitting to the Sagittarius A$^*$ data. The quantity $\alpha_c$ is a calibration factor that quantifies how accurately the ring diameter $\hat{d}_m$ tracks the shadow diameter $d_{\rm sh}$. It encompasses both theoretical and potential measurement biases and thus may be written as

$$\alpha_c = \alpha_1 \, \alpha_2 \equiv \left(\frac{d_m}{d_{sh}}\right)\left(\frac{\hat{d}_m}{d_m}\right). \tag{21}$$

Specifically, $\alpha_1$ corresponds to the ratio of the true diameter of the peak brightness of the image (bright ring) $d_m$ over the diameter of the shadow $d_{sh}$. If $\alpha_1$ equals unity, the peak emission of the ring coincides with the shadow boundary. Its value depends on the specific black-hole spacetime and the emissivity model in the surrounding plasma. A large number of time-dependent GRMHD simulations in Kerr spacetime as well as analytic plasma models in Kerr and non-Kerr metrics lead to small positive values $\alpha_1$, namely $\alpha_1 = 1 - 1.2$. This result indicates that the radius of the brightest ring is always slightly larger than the black-hole shadow.

The second calibration parameter $\alpha_2$ is the ratio between the inferred ring diameter $\hat{d}_m$ and its true value $d_m$. Three different imaging algorithms were used in the measurement of the ring diameter $\hat{d}_m$ denoted by *eht-imaging*, *SMILI* and *DIFMAP*, respectively [104]. The ring diameter was also determined by fitting analytic models, and more specifically, the *mG-ring* model [102], to the visibility data. The three imaging methods led to a value of $\alpha_2$ close to unity, while the *mG-ring* model allowed values of $\alpha_2$ in the range (1–1.3).

Employing the above, the diameter of the boundary of the black-hole shadow may be written as $d_{\rm sh} = \hat{d}_m/(\alpha_1 \alpha_2)$. Then, Equation (20) allows us to solve for the fractional

deviation $\delta$ between the inferred shadow radius $r_{\text{sh,EHT}}$ and that of a theory-specific black hole $r_{\text{sh,th}}$ [104]:

$$\delta = \frac{r_{\text{sh,EHT}}}{r_{\text{sh,th}}} - 1 \,. \tag{22}$$

The above deviation parameter allows us to test the compatibility of the EHT measurements with GR or modified theories of gravity. The posterior over $\delta$ is obtained via the formula

$$P(\delta|\hat{d}) = C \int d\alpha_1 \int d\alpha_2 \int d\theta_g \, \mathcal{L}[\hat{d}|\alpha_1, \alpha_2, \theta_g, \delta] \\ \times P(\alpha_1)P(\alpha_2)P(\theta_g)P(\delta) \,. \tag{23}$$

In the above, $\theta_g = GM/Dc^2$ is a characteristic angular size set by the black-hole mass and physical distance. Then, $\mathcal{L}[\hat{d}|\alpha_1, \alpha_2, \theta_g, \delta]$ is the likelihood of measuring a ring diameter $\hat{d}$, and $P(\theta_g)$ is prior in $\theta_g$. $P(\alpha_1)$ and $P(\alpha_2)$ are the distributions of the two calibration parameters, and $C$ a normalization constant.

To obtain the characteristic angular size $\theta_g$ of Sagittarius A$^*$, one needs its mass and distance. Two different instruments, the Keck Observatory and the Very Large Telescope, together with the interferometer GRAVITY (VLTI), were used to study the orbits of individual stars around Sagittarius A$^*$. The brightest star observed, S0-2, with a period of 16 years, has helped scientists test relativistic effects, such as gravitational redshift and the Schwarzschild precession [133–136], and to constrain alternative theories of gravity [137–139]. Its observation has also provided the most accurate measurements so far of the mass and distance of Sagittarius A$^*$. The Keck team found, for the distance, a value of $R = (7935 \pm 50 \pm 32)\,\text{pc}$ and for the black-hole mass, the value $M = (3.951 \pm 0.047) \times 10^6 \, M_\odot$ [136]. Correspondingly, the VLTI team found $R = (8277 \pm 9 \pm 33)\,\text{pc}$ and $M = (4.297 \pm 0.012 \pm 0.040) \times 10^6 \, M_\odot$. Therefore, two different priors for $\theta_g$ were derived, namely $\theta_g = 4.92 \pm 0.03 \pm 0.01 \, \mu as$ (Keck) and $\theta_g = 5.125 \pm 0.009 \pm 0.020 \, \mu as$ (VLTI).

Employing these in Equation (23), and assuming that the theory-specific solution considered in Equation (22) is the Schwarzschild solution, for which it holds $r_{\text{sh,th}} = 3\sqrt{3}\,GM/c^2 = 3\sqrt{3}\,D\,\theta_g$, the corresponding values for the deviation parameter $\delta$, along with their errors, were derived in [104] and are displayed in the first column of Table 1. We observe that the deviation $\delta$ always assumes negative values, which means that the observed black-hole shadow is found to be smaller than the one predicted by GR for the Schwarzschild black hole. We also note that the value of $\delta$ derived by employing the measurements by VLTI is consistently more negative compared to the one derived by Keck. The use of the specific algorithm for the image processing also affects the deviation parameter, with $\delta$ taking larger negative values as the *eht-imaging* algorithm is gradually replaced by the *SMILI*, the *DIFMAP* or the *mG-ring* algorithm. Finally, the value of $\delta$ is slightly modified by the type of simulations used in the calibration of $\alpha_1$; here, we employ the values obtained using the GRMHD simulations as an indicative case. We note, however, that all values derived for $\delta$ by EHT [104] are consistent with each other independently of the specific telescope, image processing algorithm or type of simulation used. For completeness, in Table 2 we present the corresponding value for the deviation parameter $\delta$ as derived by the black-hole image of M87$^*$ [126]; we observe that the central value of $\delta$ is much closer to zero, but the errors are larger due to the larger uncertainty in the measurement of the mass and distance of M87$^*$.

The definition of $\delta$ via Equation (22) in conjunction with its values in the first column of Table 1 allows us to obtain the corresponding constraints on the dimensionless quantity $r_{\text{sh}}/M$ (for notational simplicity, henceforth we drop the subscript EHT from the quantity $r_{\text{sh, EHT}}$). The 1-$\sigma$ and 2-$\sigma$ bounds on $r_{\text{sh}}/M$ are displayed in the second and third columns of Table 1 (and for completeness, in the second and third columns of Table 2). We observe that, as expected, the constraints derived from Sagittarius A$^*$ are more stringent than the

ones derived from M87$^*$: the allowed range of values in the former case is always narrower, and this leads to a consistently smaller upper limit of $r_{\rm sh}/M$.

**Table 1.** Sagittarius A$^*$ bounds on the deviation parameter $\delta$. The colored bounds are the ones we use in the plots in the main part.

| | | **Sgr $A^*$ Estimates** | | |
|---|---|---|---|---|
| | | **Deviation $\delta$** | **1-$\sigma$ Bounds** | **2-$\sigma$ Bounds** |
| *eht-img* | VLTI | $-0.08^{+0.09}_{-0.09}$ | $4.31 \leq \frac{r_{\rm sh}}{M} \leq 5.25$ | $3.85 \leq \frac{r_{\rm sh}}{M} \leq 5.72$ |
| | Keck | $-0.04^{+0.09}_{-0.10}$ | $4.47 \leq \frac{r_{\rm sh}}{M} \leq 5.46$ | $3.95 \leq \frac{r_{\rm sh}}{M} \leq 5.92$ |
| | Avg | $-0.06^{+0.064}_{-0.067}$ | $4.54 \leq \frac{r_{\rm sh}}{M} \leq 5.22$ | $4.19 \leq \frac{r_{\rm sh}}{M} \leq 5.55$ |
| *SMILI* | VLTI | $-0.10^{+0.12}_{-0.10}$ | $4.16 \leq \frac{r_{\rm sh}}{M} \leq 5.30$ | $3.64 \leq \frac{r_{\rm sh}}{M} \leq 5.92$ |
| | Keck | $-0.06^{+0.13}_{-0.10}$ | $4.36 \leq \frac{r_{\rm sh}}{M} \leq 5.56$ | $3.85 \leq \frac{r_{\rm sh}}{M} \leq 6.24$ |
| *DIFMAP* | VLTI | $-0.12^{+0.10}_{-0.08}$ | $4.16 \leq \frac{r_{\rm sh}}{M} \leq 5.09$ | $3.74 \leq \frac{r_{\rm sh}}{M} \leq 5.61$ |
| | Keck | $-0.08^{+0.09}_{-0.09}$ | $4.31 \leq \frac{r_{\rm sh}}{M} \leq 5.25$ | $3.85 \leq \frac{r_{\rm sh}}{M} \leq 5.72$ |
| *mG-ring* | VLTI | $-0.17^{+0.11}_{-0.10}$ | $3.79 \leq \frac{r_{\rm sh}}{M} \leq 4.88$ | $3.27 \leq \frac{r_{\rm sh}}{M} \leq 5.46$ |
| | Keck | $-0.13^{+0.11}_{-0.11}$ | $3.95 \leq \frac{r_{\rm sh}}{M} \leq 5.09$ | $3.38 \leq \frac{r_{\rm sh}}{M} \leq 5.66$ |
| | Avg | $-0.15^{+0.078}_{-0.074}$ | $4.03 \leq \frac{r_{\rm sh}}{M} \leq 4.82$ | $3.64 \leq \frac{r_{\rm sh}}{M} \leq 5.23$ |

**Table 2.** M87* bounds on the deviation parameter $\delta$ [126].

| | **M87$^*$ Estimates** | | |
|---|---|---|---|
| | **Deviation $\delta$** | **1-$\sigma$ Bounds** | **2-$\sigma$ Bounds** |
| EHT | $-0.01^{+0.17}_{-0.17}$ | $4.26 \leq \frac{r_{\rm sh}}{M} \leq 6.03$ | $3.38 \leq \frac{r_{\rm sh}}{M} \leq 6.91$ |

In this work, we will focus on two indicative sets of constraints, namely the ones obtained by using the *eht-imaging* method and the *mG-ring* analytic model, which lead to the smallest and largest $\delta$ (in absolute value), respectively. Moreover, in order to take a conservative stance, we will consider the Keck and VLTI values as independent and use their average value for $\delta$; these values, together with the corresponding constraints on $r_{\rm sh}/M$, are displayed in the two rows of Table 1 denoted by the word "Avg". In Sections 4–6, these mass-scale independent constraints will be used to test the viability of compact solutions arising in the context of modified gravitational theories with a scalar degree of freedom. Our analysis will pertain to current but also to future observed black-hole shadow images and will act complementary to existing works placing bounds on the parameters of these modified gravitational theories.

We would like to finish this section with the following comment. Throughout this work, we will focus on spherically symmetric solutions obtained in the context of modified theories. It is for this reason that the theory-specific solution chosen above was the Schwarzschild solution and not the Kerr one. The rotation parameter and inclination angle of Sagittarius A$^*$ does affect the observed shadow radius. However, to our knowledge, at the moment, there is no clear consensus on the value of these two parameters for Sagittarius A$^*$. In addition, it was found [128] that the shadow radius is affected very little by the rotation of the compact object, independently of the inclination angle. In fact, a recent study [140] hints toward a rather small value of $a_*$, namely $a_* \leq 0.1$. In any case, it is estimated [104] that rotating black holes can have a shadow size that is smaller than that of a non-rotating black hole by up to 7.5%. Therefore, considering the Schwarzschild solution as the theory-specific solution in our analysis seems to be a justified choice at the moment. In fact, due to the more compact geodesic structure of any rotating black hole compared

to a non-rotating one, any "Schwarzschild" constraint applied in our analysis may be considered the largest possible value for the corresponding "Kerr" one.

## 4. The Einstein-Scalar-GB Theory

We initiate our analysis by considering a scalar-tensor theory which includes a quadratic gravitational term, the Gauss–Bonnet (GB) term defined as $\mathscr{G} = R_{\mu\nu\rho\sigma}R^{\mu\nu\rho\sigma} - 4R_{\mu\nu}R^{\mu\nu} + R^2$. A general coupling function $f(\phi)$ between the scalar field $\phi$ and the GB term retains the latter—a topological invariant in four dimensions—in the theory. The action functional thus takes the following form

$$S = \frac{1}{2\kappa} \int d^4x \sqrt{-g} \left[ R - \frac{1}{2}\nabla_\alpha\phi\nabla^\alpha\phi + f(\phi)\mathscr{G} \right]. \tag{24}$$

The resulting Einstein field equations and scalar field equation, after the variation in the above action with respect to the metric tensor and scalar field, are

$$\begin{aligned}
G_{\mu\nu} = &\frac{1}{2}\partial_\mu\phi\partial_\nu\phi - \frac{1}{4}g_{\mu\nu}\partial_\rho\phi\partial^\rho\phi \\
&- \frac{1}{2}(g_{\rho\mu}g_{\lambda\nu} + g_{\lambda\mu}g_{\rho\nu})\eta^{\kappa\lambda\alpha\beta}\tilde{R}^{\rho\sigma}_{\alpha\beta}\nabla_\sigma\nabla_\kappa f(\phi),
\end{aligned} \tag{25}$$

$$\nabla^2\phi + \dot{f}(\phi)\mathscr{G} = 0, \tag{26}$$

respectively. In the second equation, the dot over the coupling function denotes its derivative with respect to the scalar field.

The EsGB theory has produced a large number of solutions describing compact objects with interesting characteristics: black holes with scalar hair [14–20,36–58], traversable wormholes [63,71,72,76] and particle-like solutions [49,86,87]. Here, we will focus mainly on the first class of solutions, namely black holes, and examine their viability under the light of the mass-scale independent constraints coming from the measurement of the shadow radius of Sagittarius A*. For the sake of comparison, we will also briefly discuss the viability of the dilatonic wormhole solutions postponing a more detailed analysis of this type of compact object for future work.

### 4.1. Black Holes

The presence of the GB term in action (24) causes the evasion of the scalar no-hair theorems and leads to the emergence of a large number of scalarized solutions, as mentioned above. In the context of the present analysis, we will consider spherically symmetric solutions that arise for three distinct coupling functions, namely for linear coupling (shift symmetry), quadratic coupling ($Z_2$ symmetry) and exponential coupling (dilatonic theory). The metric ansatz and field equations in explicit form may be found in Appendix A. For the details on constructing these solutions, the interested reader may consult, for instance, the works [17,45,46,141,142].

In principle, our solutions require the specification of three parameters beyond GR. We first need to specify the coupling constant $\alpha$, which quantifies the strength of the interaction between the Gauss–Bonnet curvature invariant and the scalar field; we also need two boundary conditions for the scalar field since it obeys a second-order differential equation. Assuming a simple Taylor expansion of the scalar field around the horizon $\phi(r) = \phi_h + \phi_{h,1}(r - r_h) + ...$, it has been shown in several works (see, for instance, [17,45]) that one may only obtain solutions with a regular horizon as long as the following constraint holds

$$\phi_{h,1} = -\frac{r_h}{4\dot{f}_h}\left(1 \mp \sqrt{1 - \frac{96}{r_h^4}\dot{f}_h^2}\right). \tag{27}$$

This reduces the parameters from three to two, namely the field value at the horizon $\phi_h$ and the coupling strength $\alpha$. In addition to the preceding constraint, we also need to limit the

two-dimensional plane $(\phi_h, \alpha)$ due to the requirement that the quantity under the square root in (27) is positive definite. For this reason, we will trade parameter $\alpha$ with $\beta$, defined as follows

$$\beta \equiv \frac{\sqrt{96}}{r_h^2} \dot{f}_h. \tag{28}$$

In this way, the parameter space we need to scan is $(\phi_h, \beta)$ with $-1 < \beta < 1$ defined within clear boundaries. After the study of the complete parameter space, our results will be eventually expressed again in terms of $\alpha$ for clarity.

### 4.1.1. $f(\phi) = \alpha \, \phi(r)$

First, we consider the linear coupling scenario, which yields a shift-symmetric term, considering the fact that the Gauss–Bonnet invariant is a total divergence in four dimensions. Shift symmetry prevents the scalar from acquiring a mass that would lead to an exponential suppression of GR deviations in the strong-field regime of gravity. Therefore, scalar fields respecting shift symmetry are particularly motivated. For the case of linear coupling, the two-dimensional parameter space $(\phi_h, \beta)$ described above is reduced to one-dimensional parameter space since the value of the field does not enter in the field equations as a result of the shift symmetry. In that case, the solutions are expected to form a line in the $(\alpha/M^2, r_{sh}/M)$ plane that spans the $-1 < \beta < 1$ parameter range.

This is indeed the case, as seen in Figure 2, where we depict the rescaled black-hole shadow $r_{sh}/M$ in terms of the dimensionless parameter $\alpha/M^2$ of the theory. We always choose positive values of $\phi$ so that the sign of the coupling parameter $\alpha$ directly reflects the sign of $\beta$. Here, we consider both positive and negative values for the coupling parameter and present the complete family of solutions for the allowed range $-1 < \beta < 1$. As $|\beta| \to 1$, the minimum mass solutions—a characteristic feature of the EsGB theory—are reached, and the solution lines terminate. We also note that the line of the solutions is mirror symmetric around the $\alpha = 0$ line. This is expected since the field equations, as well as the initial conditions, are symmetric under the simultaneous exchange of the sign of $\dot{f} = \alpha$ and $(\phi', \phi'')$. This, of course, only holds for the linear coupling, for which $\ddot{f} = 0$.

We should note here that the aforementioned mass parameter $M$ of the black hole is the one that appears in the coefficient of the $1/r$ term in the far-field expansion of the metric component $g_{00}$. In a modified gravitational theory, this parameter does not necessarily coincide with the mass obtained using the ADM [143] or field-theoretic formalism [112,144–146]. It may also differ from the black-hole mass calculated by considering the orbital motion of a test body around the black hole. All these notions of mass, although trivially related in the context of General Relativity, may indeed be different in the framework of a generalized theory of gravity. However, in all modified theories considered in this work, the presence of additional fields does not affect the coefficient of the $1/r$ term in the far-field expansion of the metric component $g_{00}$. For instance, in the Einstein-scalar-GB theory that we study here, the scalar charge appears in the $1/r^3$ term at the earliest in the expansion of $g_{00}$, while the scalar coupling function appears in the $1/r^4$ term. As a result, we do not anticipate any significant differences between parameter $M$ determined through the asymptotic expansion of the metric and the ADM mass or the conserved Noether charge. The black-hole mass calculated via the orbital motion of a test body is also not expected to be different from parameter $M$ as long as the radius of the orbit does not lie very close to the black-hole horizon.

Coming back to Figure 2, we observe that the shadow radius $r_{sh}/M$ decreases as $\alpha/M^2$ increases. This is easily understood if we recall (see, for example, [17,45,46,131]) that the GB term causes a negative contribution to the total energy density of the theory and thus exerts a repulsive force. Therefore, if a black hole is to be created, any matter distribution needs to be compacted into a smaller area of spacetime compared to the case where the GB term is absent. As a result, the GB scalarized black holes always have a smaller horizon radius than, e.g., the Schwarzschild black hole with the same mass [45,46,131]. Since the whole geodesic structure gets more compact as $\alpha$ increases, the shadow radius will also get smaller. This decreasing trend of the solution line holds for both $\alpha > 0$ and $\alpha < 0$ since the

GB contribution to the energy density is negative independently of the sign of $\alpha$. In fact, it is proportional to the combination $\phi_{h,1}\dot{f}_h$, which is always negative according to Equation (27). This also holds independently of the exact form of the coupling function $f(\phi)$, and thus we expect to see a similar behavior for the other two forms of $f$. The generically smaller size of the shadow radius of any EsGB black-hole solution compared to the GR one brings to the foreground these types of modified theories since the EHT constraints [104] point to observed shadow radii, which are always smaller than the Schwarzschild one.

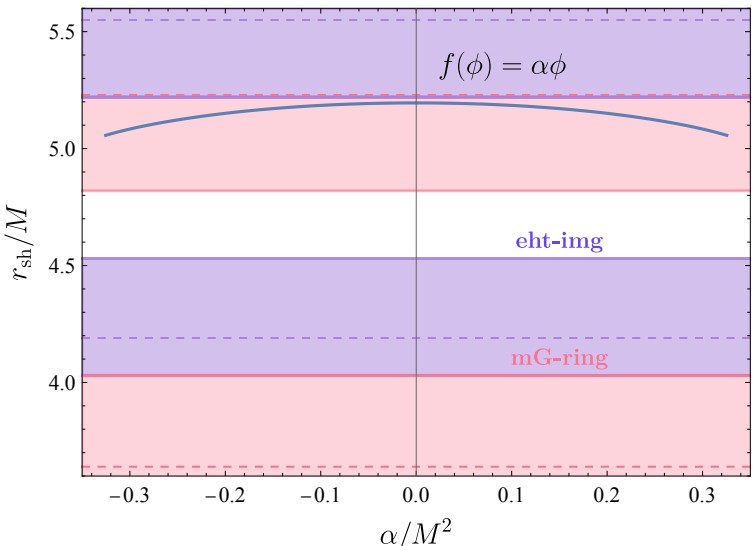

**Figure 2.** Shadow radius for EsGB theory with linear coupling. The blue line scans the full range of values of $\beta$ defined in (28) with the left and right endpoints corresponding to $\beta = -1$ and $\beta = 1$, respectively. The horizontal solid and dashed lines denote the EHT 1-$\sigma$ and 2-$\sigma$ allowed ranges, respectively; the blue lines correspond to the *mG-ring* bound and the red lines to the *eht-imaging* bound.

Let us now focus on the constraints imposed on the shift symmetric theory by the mass-scale independent bounds depicted in Table 1. As explained in Section 2, we will employ two of the derived bounds: the most 'conservative' bound, the *eht-imaging* one, which yields the smallest central value of the fractional deviation $\delta$, and the most 'liberal' bound, the *mG-ring* one, which allows for larger deviations from GR. The two solid, horizontal, blue lines denote the allowed 1-$\sigma$ range by the *eht-imaging* bound, while the two solid, horizontal, red lines denote the corresponding range allowed by the *mG-ring* bound (the blue and red horizontal, dashed lines denote the corresponding 2-$\sigma$ bounds). Likewise, the blue-shaded area is the one excluded by the *eht-imaging* bound within 1-$\sigma$ accuracy and the red-shaded area is the one excluded by the *mG-ring* bound. The white area is the one that is allowed by both bounds.

According to Figure 2, the complete range of scalarized solutions in the shift symmetric EsGB theory is compatible with the *eht-imaging* bound, while it is altogether excluded by the *mG-ring* bound within 1-$\sigma$! Our findings highlight in the best possible way the need to 'bridge the gap' between the different EHT bounds as they lead to conflicting conclusions regarding the viability of certain solutions and, in a more general context, the physical relevance of their underlying theories. We note that all solutions found, which are allowed by the *eht-imaging* bound, also satisfy the recent experimental constraint on the dimensionless parameter $\alpha/M^2 < 0.54$ [147] set on shift symmetric EsGB theories by the detection of gravitational waves from black-hole binaries. If, on the other hand, one takes a more conservative approach and demands that viable solutions should satisfy both of the EHT bounds within 1-$\sigma$ accuracy, one is forced to exclude the complete range of scalarized, shift-symmetric solutions as none of them falls in the optimum white area of the plot. All solutions are still allowed within 2-$\sigma$ accuracy.

### 4.1.2. $f(\phi) = \frac{\alpha}{2} \phi(r)^2$

The quadratic coupling function poses particular interest in the context of spontaneous scalarization, which will be analyzed in the following sections. Unlike the linear case, the case of the quadratic coupling function necessitates searching along a two-dimensional parameter space due to the fact that the initial value of the field $\phi_h$ is physical. In order to facilitate the search, we select N = 25 points equally spaced in the $\ln(\phi_h)$ space with $\phi_{h,\min} = 0.1$ and $\phi_{h,\max} = 100$. For each of these N points, i.e., for each choice of $\phi_h$, we plot a line that spans the entire parameter range $-1 < \beta < 1$. The results are displayed in Figure 3. The red dots in the figure denote a transitioning point regarding the sign of $T_r^{r\prime}$ near the horizon, which will be discussed shortly.

The lines in Figure 3 denote solutions where large $\phi_h$ are generally consolidated close to the vertical axis. In contrast, the smaller $\phi_h$ is, the more the lines spread out to larger values of $|\alpha/M^2|$. This is expected due to the definition of beta, which, in this case, takes the form

$$\beta = \frac{\sqrt{96}}{r_h^2} \alpha \, \phi_h \,. \tag{29}$$

It is clear that in order to reach the values of $\beta \approx \pm 1$, i.e., the limits of the range of $\beta$, we need to choose an increasingly larger $\alpha$ in order to compensate for the smallness of $\phi_h$. This justifies the fact that the lines extend further and further away from the origin for small $\phi_h$ values.

Additionally, one may readily observe that the symmetry under the change in the sign of $\alpha$, present in the shift-symmetric case, is now broken. In fact, for negative values of the coupling constant $\alpha$, both the mass parameter $M$ and the shadow radius $r_{\rm sh}$ are affected much more dramatically compared to the positive coupling case. This is manifest in Figure 3, where each solution line for $\alpha < 0$ extends along a larger range of values of $r_{\rm sh}/M$ compared to that of the $\alpha > 0$ solutions. The fact that the former lines turn downward and to the right comes as a consequence of the dimensionless normalization we have applied to the axes. In addition to that, we have numerically observed that negative values of the coupling $\alpha$ lead to large and negative values of the scalar charge. The largeness of the charge and mass for values of the coupling deep into the negative regime is a generic consequence of the evolution of the field equations at intermediate scales between the horizon and infinity, and hence it is difficult to understand the origin of this effect by studying the asymptotic behavior of the solutions.

Moreover, it is interesting to note that, for some of the parameter space analyzed, one crosses the boundary beyond which one can obtain a solution with $\lim_{r \to r_h} T_r^{r\prime}(r) > 0$. We remind the reader that the condition $\lim_{r \to r_h} T_r^{r\prime}(r) < 0$, satisfied by the scalarized solutions found in [17,45,46], was employed to demonstrate the violation of the novel no-hair theorem [35]. Using the results of [142], we can compute the boundary beyond which solutions with $\lim_{r \to r_h} T_r^{r\prime}(r) > 0$ appear as follows

$$\lim_{r \to r_h} T_r^{r\prime}(r) = 0 \quad \Rightarrow \quad \alpha = -\frac{1}{4} + \frac{\beta^2}{16} - \frac{2\sqrt{1-\beta^2}}{9} \,. \tag{30}$$

Simultaneously, due to the definition of $\beta$, we can write

$$\alpha = \frac{r_h^2}{4\sqrt{6}\phi_h} \beta \,. \tag{31}$$

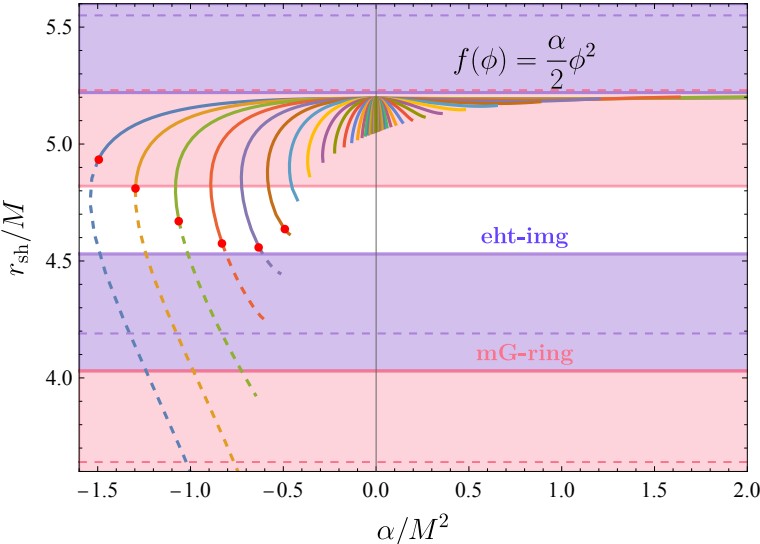

**Figure 3.** Shadow radius for EsGB theory with a quadratic coupling. Each colorful line scans the full range of parameter $\beta$ for a different fixed value of $\phi_h$. The endpoint of the lines in the negative and positive regime of the horizontal axis correspond to $\beta = -1$ and $\beta = 1$, respectively. The red dots denote the point in the parameter space at which condition (30) is satisfied. The horizontal solid and dashed lines denote the EHT bounds as before.

The above result implies that depending on the choice of $\phi_h$ and $\beta$, $\alpha$ can be above or below the boundary defined by (30). The points below the boundary, i.e., scalarized black-hole solutions with $\lim_{r \to r_h} T_r^{r\prime}(r) > 0$, are denoted by dashed lines in Figure 3, and the transitioning points are marked by large red dots. We note that such solutions arise only in the case of negative coupling constant $\alpha$, and thus any analyses considering only positive $\alpha$ are bound to overlook them.

Figure 3 leads to similar conclusions regarding the validity of the quadratic, scalarized GB solutions with positive $\alpha$ to the ones found for the linear-coupling case: the *eht-imaging* bound allows the complete range of solutions while the *mG-ring* bound excludes all of them within 1-$\sigma$! No scalarized solutions with positive $\alpha$ fall in the white area. However, the situation is radically different for solutions with negative $\alpha$. There, as noted above, the lines of solutions with small or intermediate values of $\phi_h$ extend into the white area and thus survive all EHT bounds. These favored solutions are characterized by either a positive or negative value of $\lim_{r \to r_h} T_r^{r\prime}(r) > 0$.

### 4.1.3. $f(\phi) = \alpha\, e^{\gamma\, \phi(r)}$

We will finally consider the exponential coupling function, which, as has been pointed out in various previous works (e.g., see [17,18]), is motivated by the low-energy limit of heterotic superstring theory [148]. For the dilatonic coupling, we need to scan a three-dimensional parameter space since there is an additional parameter $\gamma$ that characterizes the coupling function. We follow the same procedure as before and display the results for two distinct values of $\gamma = 1, 2$ in Figures 4 and 5, respectively.

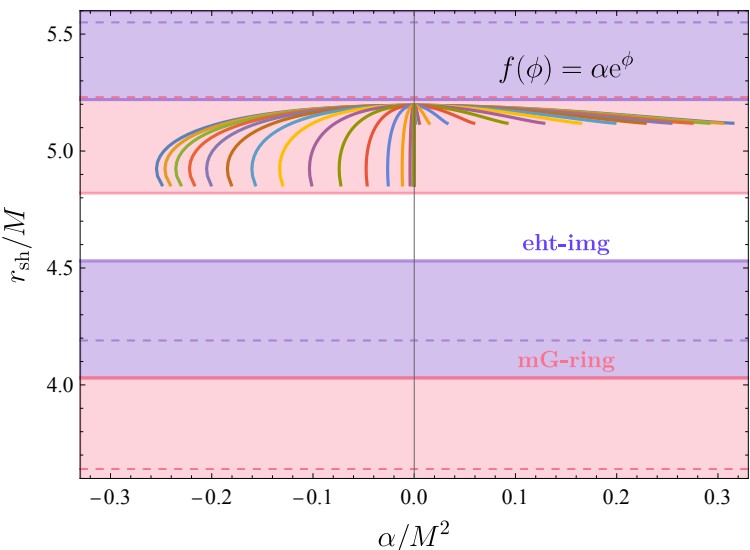

**Figure 4.** Shadow radius for EsGB theory with a dilatonic coupling with $\gamma = 1$. The colored lines have the same meaning as in Figure 3, while the horizontal solid and dashed lines denote the EHT bounds as before.

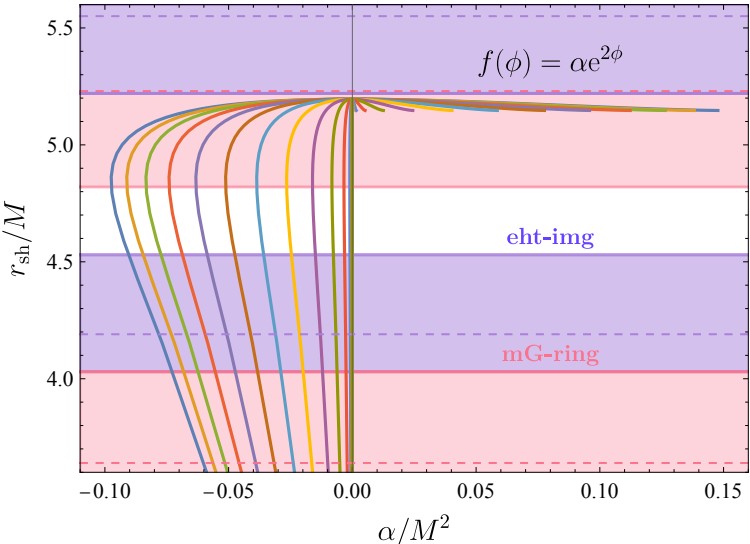

**Figure 5.** Shadow radius for EsGB theory with a dilatonic coupling with $\gamma = 2$. The colored lines have the same meaning as in Figure 3, while the horizontal solid and dashed lines denote the EHT bounds as before.

The subclass of solutions derived for positive values of the coupling parameter $\alpha$ exhibit the same profile, for both values of $\gamma$, as in the previous two cases: the whole range of solutions extends over a very restricted range of values of $r_{\rm sh}/M$. As a result, they are all allowed by the *eht-imaging* bound, but they are all also excluded by the *mG-ring* bound within 1-$\sigma$. No positive-$\alpha$ solution manages to satisfy both bounds. In fact, all GB scalarized black holes derived for positive $\alpha$ demonstrate the same profile when it comes to their viability under the Sagittarius A* constraints independently of the particular form of the coupling function $f(\phi)$. We note, however, that all these solutions satisfy the theoretical bound $\alpha/M^2 < 0.69$ for the existence of scalarized dilatonic black holes [17,37], and the experimental bound $\alpha/M^2 < 0.54$ [147] (which although was derived for the shift

symmetric case may also apply in the exponential case in the limit of small $\alpha$ as is the case here).

The situation, however, is different when we consider the solutions derived for negative values of the coupling constant $\alpha$. Considering the behavior observed in the previous two cases as well as the one depicted in Figures 4 and 5, we conclude that this subclass of solutions is affected both by the form of the coupling function $f(\phi)$ and the particular values assumed for the parameters of the theory. In Figure 4, we see that, for $\gamma = 1$, none of the negative-$\alpha$ solutions manages to satisfy both EHT bounds despite the fact that they extend over a larger range of values of $r_{\rm sh}/M$ compared to the positive $\alpha$ solutions. However, for $\gamma = 2$, the solution lines manage to extend across the white optimum area and thus, a subgroup of solutions for a specific range of $\alpha$ are favored by the EHT constraints and may be rendered viable.

We note that the only way to cross into the regime with $\lim_{r \to r_h} T_r^{r\prime}(r) > 0$ for the dilatonic coupling is to increase the value of $\gamma$ even further. However, this yields a less observationally motivated theory. Another important observation is that for the dilatonic coupling, the ratio $r_{\rm sh}/M$ depends on $\gamma$ but not on $\alpha$ and $\phi_h$ simultaneously—this is reflected in the fact that all solution lines corresponding to different values of $\phi_h$ terminate at the same horizontal line in Figures 4 and 5. This is due to the presence of a symmetry in the Lagrangian that allows us to absorb any change in the value of the scalar field on the horizon $\phi_h$ into a redefinition of the coupling constant $\alpha$ [17], namely

$$\phi_h \to \phi_h + \phi_* , \quad \alpha \to \alpha\, e^{-\gamma \phi_*} . \tag{32}$$

As a result, the parameter space reduces from a three-dimensional to a two-dimensional one.

*4.2. Wormholes*

In the context of theory (24), traversable wormhole solutions have been discovered for a variety of scalar-GB couplings featuring single- or double-throat geometries [71,72,131]. Exploring these solutions in depth is beyond the scope of this work and is left for future analysis. Here, however, we will present the results for one characteristic example in order to demonstrate the potential of our analysis as a tool to observationally distinguish wormhole from black-hole solutions.

The case we consider here is the first one historically studied [71,72] and involves an exponential coupling function of form $f(\phi) = \alpha e^{-\gamma \phi}$ with $\gamma = 1$. Single-throat solutions are then discovered if one assumes the line element given in (10). In accordance with the black-hole scenario, a regularity for the scalar field's derivative on the throat is derived

$$\phi_0^{\prime 2} = \frac{f_0(f_0 - 1)}{2\alpha e^{-\phi_0}\left[f_0 - 2(f_0 - 1)\frac{\alpha}{l_0^2}e^{-\phi_0}\right]} , \tag{33}$$

where $f_0$ and $\phi_0$ are the values of $f$ and $\phi$ evaluated at the throat. For simplicity, we chose $\phi_0$ so that, asymptotically, the field vanishes. Additionally, the value of the other metric function $v_0$ at the throat is chosen so that an asymptotically flat spacetime is recovered. We are left, therefore, with one free parameter, i.e., $f_0$, in addition to the coupling one.

In the limit $f_0 \to 1$, the redshift function $v_0$ tends to larger negative values, and a horizon emerges, thus yielding the relevant black-hole solutions in this theory. This allows us to directly compare the shadow radii between black holes and wormholes arising for $\gamma = 1$. The results are presented in Figure 6, where we see that $f_0$ has non-trivial consequences both on the shadow radius and on the mass range of the solutions. Specifically, it appears that as we increase $f_0$ the mass range can also increase significantly. In terms of the shadow radius, we see that all solutions—including the black hole—presented lay within the averaged 1-$\sigma$ *eht-imaging* bounds presented in Table 1. On the other hand, all solutions are excluded within 1-$\sigma$ if one chooses to consider the averaged *mG-ring* estimates. Once again, no solution exists that satisfies both bounds.

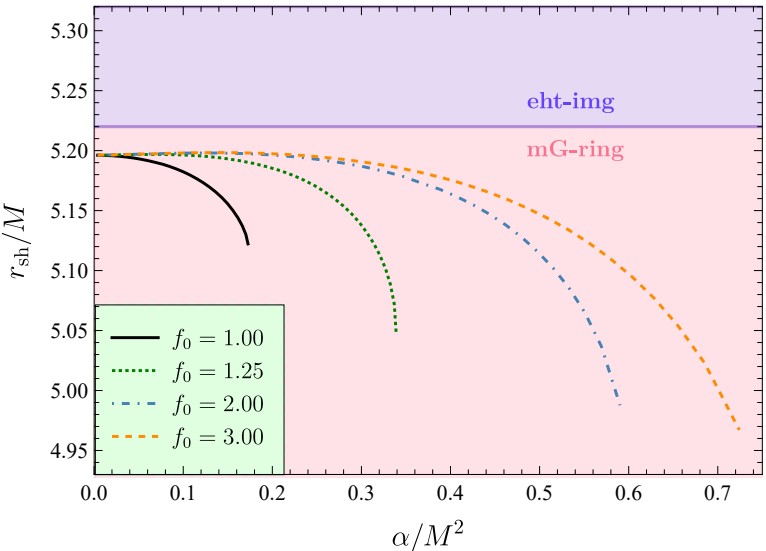

**Figure 6.** Wormhole solutions in EsGB theory with coupling function $f(\phi) = \alpha e^{-\phi}$, for $f_0 = \{1, 1.25, 1.5, 2, 3\}$.

## 5. Curvature-Induced Spontaneous Scalarization

A particular class of scalar-tensor theories, in the more general framework of Horndeski theory, has attracted a lot of attention and has been extensively scrutinized over recent years. This class pertains to a phenomenon known as *spontaneous scalarization* of compact objects (black holes and neutron stars). It describes solutions *spontaneously* endowed with scalar hair as a consequence of a "phase transition" associated with the emergence of a tachyonic instability. Beyond a certain compactness threshold, black holes tend to transition from unstable, unscalarized GR solutions to stable scalarized configurations. The main reason why this particular class of theories entails exceptional interest relates to the fact that GR is retrieved in the weak gravitational-field regime, while deviations are only detected in heavily curved spacetimes.

The initially theorized model [47,48] considered GR supplemented by a kinetic term for the scalar field plus a non-minimal interaction of the scalar field with the GB invariant. For spontaneous scalarization to be realized, it is crucial that this coupling satisfies a certain number of conditions, which will be discussed in the following paragraphs. However, this initial model has been shown to be unstable under radial perturbations [149,150].

Following arguments discussed in detail in [151], we can write a general action allowing for spontaneously scalarized solutions to emerge in the following form:

$$S = \frac{1}{2\kappa} \int \mathrm{d}^4x \sqrt{-g} \left[ R - \frac{1}{2} \nabla_\alpha \phi \nabla^\alpha \phi + h(\phi)R + f(\phi)\mathscr{G} + V(\phi) \right], \tag{34}$$

where $\mathscr{G}$ is again the Gauss–Bonnet invariant. The scalar field self-interactions have been shown to non-trivially affect the properties of the scalarized solutions. This includes the threshold of scalarization that is altered by the bare mass term and the radial stability that is improved if one includes quartic interactions [54]. Here, we consider $V(\phi) = 0$ for simplicity. Another way to stabilize black-hole solutions in this theory is to include higher-order operators in the GB coupling function $f \sim \alpha\phi^2 + \zeta\phi^4$. Provided that $\zeta$ is a large enough negative multiple of $\alpha$, solutions can indeed be stabilized [152].

The field equations for the metric that one derives by varying action (34) are:

$$G_{\mu\nu} = T^\phi_{\mu\nu}, \tag{35}$$

where the scalar-field energy-momentum tensor is given by:

$$
\begin{aligned}
T_{\mu\nu}^{\phi} =& \frac{1}{2}\nabla_{\mu}\phi\nabla_{\nu}\phi - \frac{1}{4}g_{\mu\nu}(\nabla\phi)^2 - \left(g_{\mu\nu}\nabla^2 - \nabla_{\mu}\nabla_{\nu}\right)h(\phi) \\
& - h(\phi)G_{\mu\nu} - \frac{1}{g}g_{\mu(\rho}g_{\sigma)\nu}\epsilon^{\kappa\rho\alpha\beta}\epsilon^{\sigma\gamma\lambda\tau}R_{\lambda\tau\alpha\beta}\nabla_{\gamma}\nabla_{\kappa}f(\phi).
\end{aligned}
\tag{36}
$$

Since, in this work, we assume spherical symmetry, we must recover the Schwarzschild geometry asymptotically. Perturbing the scalar equation around the GR solution ($\phi = \phi_0 + \delta\phi$), we find:

$$
\Box\phi = -\left[\dot{f}(\phi)\mathscr{G} + \dot{h}(\phi)R\right] \Rightarrow \Box\delta\phi = -\ddot{f}(\phi_0)\mathscr{G}\delta\phi.
\tag{37}
$$

The term $-\ddot{f}(\phi_0)\,GB$ acts as an effective mass for the scalar field; therefore, when it becomes significantly negative, it triggers a tachyonic instability. The first spontaneous scalarization condition, therefore, requires $\ddot{f}(\phi_0)\mathscr{G} > 0 \Rightarrow \ddot{f}(\phi_0) > 0$ since $\mathscr{G} > 0$ in the exterior of spherically symmetric black holes. If we also integrate the scalar equation by parts, it is straightforward to show that for spontaneously scalarized black holes to emerge; it is also required that $\phi\dot{f}(\phi) > 0$. This second condition that the coupling function should satisfy relates to GR being included in this framework, i.e., $f(\phi_0) = 0$ for some $\phi_0$.

The scalarization occurs beyond a threshold mass, which is found by examining the linear stability of scalar perturbations around the Schwarzschild background. To that extent, the scalar perturbation is decomposed as follows

$$
\delta\phi = \frac{\sigma(r)}{r}Y_{\ell}^{m}(\theta,\phi)\,e^{-i\omega t},
\tag{38}
$$

where $Y_{\ell}^{m}(\theta,\phi)$ are the spherical harmonics. For spherical symmetry, the above yields an equation of the following type:

$$
\frac{d^2\sigma}{dr^2} + \omega^2\sigma = V_{\text{eff}}(\ell,\alpha)\,\sigma.
\tag{39}
$$

The effective potential depends on the theory, and $\alpha$ corresponds to the coupling parameter appearing within $f(\phi)$. Requiring the existence of bound solutions to the above equation that satisfy the proper asymptotic properties (equivalence with square integrability in quantum mechanics) allows us to determine the discrete spectrum of scalarization thresholds depending on the mode $n$ and the angular number $l$. For a massless scalar field with $\ell = 0$, the thresholds for the fundamental mode and the first overtone are found to be $\hat{M}_{\text{th}}^{(0)} \approx 1.179$ and $\hat{M}_{\text{th}}^{(1)} \approx 0.453$, respectively. Here, and in what follows, we have defined the dimensionless mass parameter

$$
\hat{M} = M_{\text{ADM}}/\alpha^{1/2},
\tag{40}
$$

where $M_{\text{ADM}}$ is the ADM mass of the solution, which is read off the asymptotic expression of $g_{rr}$. This is performed so that our results are directly comparable with the existing bibliography.

In the next two subsections, we address two particular models, the *minimal model* characterized by coupling functions of quadratic form and the *quartic sGB model*, where the coupling function to the GB term has been supplemented by a quartic function of the scalar field.

### 5.1. Minimal Model

Here, we consider the minimal model associated with spontaneous scalarization identified in [151] and explored in [150,153–156], where the coupling functions are defined as

$$h(\phi) = -\frac{\beta}{2}\phi^2 \quad , \quad f(\phi) = \frac{\alpha}{2}\phi^2 , \tag{41}$$

where $\beta$ and $\alpha$ are coupling constants. This subclass of scalarization models has a particular interest as it addresses a number of issues traditionally associated with scalarization. Specifically: (i) it suppresses neutron-star scalarization leading to avoidance of binary pulsar constraints [155], (ii) it allows for a late-time cosmological attractor to GR [154], (iii) it yields stable scalarized black-hole solutions [150,156] and (iv) it improves the hyperbolicity of the formulation [150]. Considering the various benefits of this sRGB synergy, here we try to test its implications to black-hole shadows.

To this end, in Figure 7, we present the shadow radius for spontaneously scalarized black-hole solutions derived for different values of the scalar-Ricci coupling constant $\beta$. In terms of cosmological consistency, it has been pointed out that negative $\beta$ values require substantial fine-tuning if one wants to retrieve a late-time attractor. This fine-tuning, however, is not required when $\beta > 0$, when a GR attractor is naturally recovered at late times. Therefore, we will be considering only positive values for $\beta$ in what follows. Positive values of $\beta$ have also been shown to improve the hyperbolic formulation of the scalar perturbations equation [150]. Further, changing the value of $\beta$ has been shown to change the gradient of the curves in the scalar charge-mass plots, a relation directly associated with the stability of the solutions [152,156]. A positive/negative gradient describes unstable/stable solutions. In general, we can distinguish between three regions: (I) $\beta \lesssim 1$ solutions are unstable, (II) $1 \lesssim \beta \lesssim 1.2$ the solution curves have both a stable and an unstable part (effectively yielding one stable and one unstable solution for any $\hat{M}$) and (III) $\beta > \beta_{\text{crit}} \approx 1.2$ when all solutions are stable. Finally, values of $\beta$ close to one achieve scalarization suppression for neutron stars [155] and avoid significantly influencing the formation of large-scale structures.

However, here we aim to conduct a comprehensive study and thus, in Figure 7, we present the results for the radius of the black-hole shadow in the minimal model for a variety of values of $\beta$, namely $\beta = \{0, 5, 10, 50, 100\}$. The left panel depicts the solutions for the fundamental mode ($n = 0$). Here, case $\beta = 0$, shown with a solid red line, corresponds to the radially unstable sGB scalarization model. The solutions, in this case, lie to the right of the scalarization threshold at $\hat{M}_{\text{th}}^{(0)} \approx 1.179$. The rest of the curves shown correspond to values of $\beta$ that are larger than the critical value and, therefore, to stable configurations. The right panel shows the solutions for the first overtone ($n = 1$). Here, only the solutions with $\beta \gtrsim 10$, which lie to the left of the threshold scalarization value of $\hat{M}_{\text{th}}^{(0)} \approx 0.453$, are stable. In both plots, the horizontal axis depicts the value of the dimensionless parameter $\hat{M}$ defined in Equation (40). The vertical axis showing the shadow radius $r_{\text{sh}}$ of the black hole is also properly rescaled in terms of the mass $M$ so that the results are independent of the black-hole mass under consideration.

We readily observe that significant deviations from GR appear in the value of the shadow radius, especially toward the lower mass limit of each curve. This is to be expected since it is for the lightest black holes that the curvature is stronger, and the effect of both the GB and additional Ricci term becomes increasingly more important. As in the EsGB theory, the quadratic GB term leads to black holes with a more compact geodesic structure, compared to the Schwarzschild solution with the same mass, with the radius of the black-hole shadow following along and taking smaller values, too. The more conservative *eht-imaging* bound allows all of the solutions to 1-$\sigma$ accuracy, whereas the *mG-ring* bound, which favors larger deviations from GR, excludes almost all of the solutions to 1-$\sigma$ accuracy. The only solutions allowed are the ones toward the bottom tip of the curves for the fundamental modes. Considering the mG-ring 2-$\sigma$ bounds, however, all solutions are allowed.

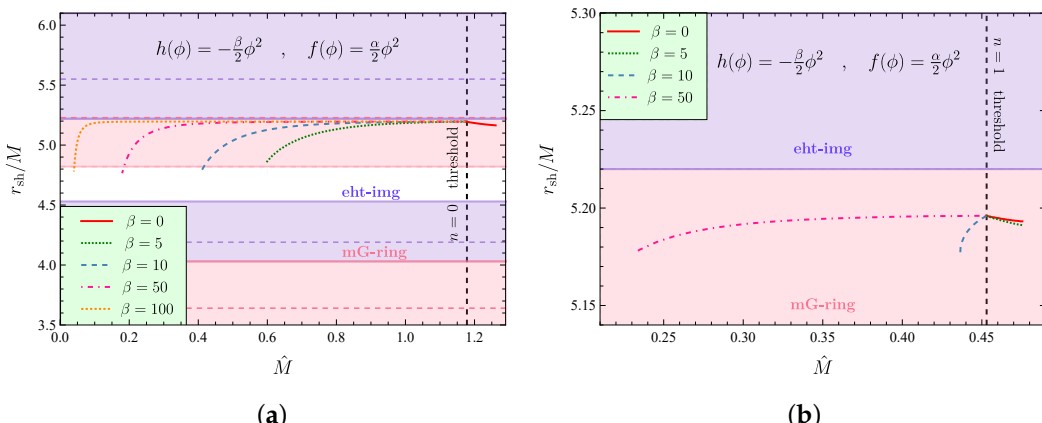

(**a**)  (**b**)

**Figure 7.** (**a**) Shadow radius of the fundamental mode ($n = 0$) for spontaneously scalarized black holes in the EsRGB theory with quadratic couplings between the scalar field and curvature. The values of $\phi$-$R$ coupling for the lines plotted are $\beta = 0, 5, 10, 50, 100$. At the same time, the $\phi$-$\mathscr{G}$ coupling spans all the allowed values for which spontaneously scalarized solutions are retrieved. (**b**) Same as left panel but for the first overtone $n = 1$. The $\beta = 100$ case is not presented here for illustrative purposes as it extends to values of $\hat{M}$ that are much smaller than the rest.

Therefore, if future observations of horizon-scale images of much lighter black holes are made with the same error bounds, scalarized black-hole solutions would be either favored or even admitted as the only possible choice compared to the GR solution. Focusing on the character of Sagittarius A*, though, spontaneous scalarization may not be a viable option: all stable solutions arise in the regime $\hat{M} < 1.2$, which translates to $0.7 < \alpha/M^2$. If, in addition, we focus on the subclass of solutions, which survive both the *eht-imaging* and the *mG-ring* bounds, these emerge for $\beta \gtrsim 7$ in the regime $\hat{M} < 0.5$ or for $4 < \alpha/M^2$. At the moment, there are no bounds on the dimensionless parameter $\zeta \equiv \alpha/M^2$ derived in the context of the EsRGB theory. However, if we take the theoretical bound $\zeta < 0.69$ for the existence of dilatonic black holes [17,37] or the experimental bound $\zeta < 0.54$ [147] for shift symmetric solutions as indicative values, we see that the aforementioned range significantly surpasses the latter ones. A more detailed study dedicated to the EsRGB theory needs to be performed before concluding whether Sagittarius A* is a spontaneously scalarized black-hole solution.

*5.2. Quartic sGB Coupling*

Here we examine a variation in the EsGB model (without the Ricci coupling) that has been shown to yield stable black-hole solutions under certain assumptions [152]:

$$h(\phi) = 0 \quad , \quad f(\phi) = \frac{\alpha}{2}\phi^2 + \frac{\zeta}{4}\phi^4. \tag{42}$$

As mentioned earlier, for sufficiently negative values of the ratio $\zeta/\alpha \lesssim -0.7$, black holes do become stabilized. Considering positive ratios, on the other hand, produces solutions that are unstable. As in the minimal model, there is a particular range of negative values for the ratio $\zeta/\alpha$ for which both stable and unstable solutions emerge.

One of the reasons why this model is particularly interesting relates to the fact that even for small values of the quartic coupling, the minimum mass can, in principle, be pushed to very small values, contrary to the minimal model presented in the last subsection. This feature has evaded attention in other works and is of significant importance as it allows us to probe a much larger range of masses. A consequence of this large mass range is an equally large range in the shadow radii, as can be seen in Figure 8. It is important to mention that the minimal mass for any $\zeta/\alpha \lesssim -0.7$ seems to have the potential to be arbitrarily pushed to small values.

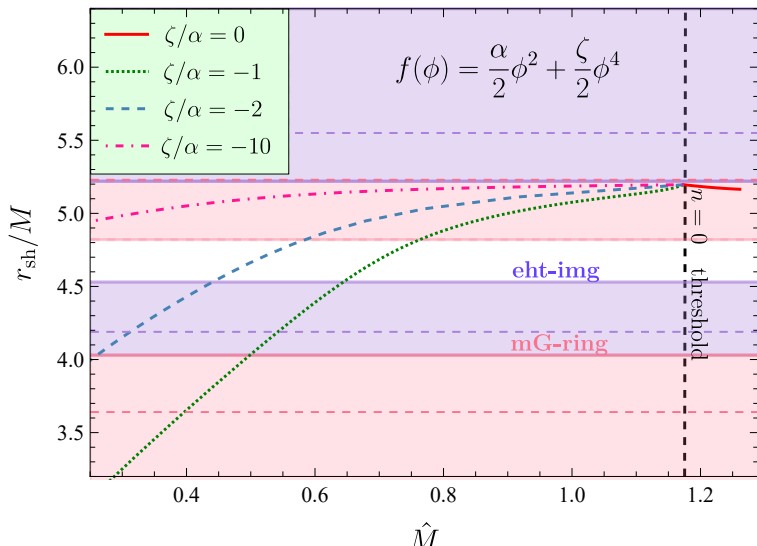

**Figure 8.** Shadow radius of the fundamental modes ($n = 0$) for spontaneously scalarized black holes in EsGB theory with a quartic $\phi$-GB coupling, for different ratios $\alpha/\zeta = \{0, -1, -2, -10\}$.

Employing the mass-scale independent bounds of Table 1, we may draw a number of useful conclusions. To start with, solutions with fairly large, negative values of $\zeta/\alpha$, i.e., $\zeta/\alpha \simeq -10$, seem to be excluded by the *mG-ring* bound, at least in the intermediate and larger mass regime. For less negative values of $\zeta/\alpha$ the region allowed by the bounds from Table 1 is pushed to intermediate masses. In general, for some fixed $\zeta/\alpha$, solutions with large masses tend to be disfavored by the *mG-ring* bound, while small-mass solutions are excluded by the *eht-imaging* bound, and this holds independently of the value of that ratio.

We note that, in this case, the solutions that are allowed by the existing bounds of Table 1 emerge for $\hat{M} < 0.85$ or for $1.4 < \alpha/M^2$. This is an improvement since the lower bound on $\alpha/M^2$ is now much closer to the indicative theoretical and experimental bounds mentioned earlier. Again, in the absence of a bound on $\alpha/M^2$, specifically for the quartic EsGB model, we cannot conclusively state whether Sagittarius A* can be a spontaneously scalarized solution arising in the framework of this model.

## 6. The Einstein–Maxwell-Scalar Theory

In the black-hole scenario, there exists a wider class of theories that also includes Einstein–Maxwell-Scalar (EMS) models as spontaneous-scalarization frameworks [157]. The EMS model describes a scalar field non-minimally coupled to Maxwell's tensor while being minimally coupled to gravity. It has been shown that under certain assumptions, black-hole solutions appear to spontaneously scalarize [157–159]. For small values of charge to mass ratio $q$, these solutions have been demonstrated to be the endpoints of dynamical evolution of unstable Reissner–Nordström (RN) solutions with the same $q$ within numerical error, while for larger values, dynamical scalarization decreases its value. The action functional describing the EMS theory is given by:

$$S = \frac{1}{2\kappa} \int d^4x \sqrt{-g} \left[ R - \frac{1}{2} \nabla_\alpha \phi \nabla^\alpha \phi + f(\phi) F_{\mu\nu} F^{\mu\nu} \right].$$ (43)

The theory we consider here admits the RN solution, which is scalar-free. To accommodate this, we require that asymptotically our theory must approach the RN solution, which translates to $\phi \to 0$ and $f(\phi) \to -1$, as $r \to \infty$.

The Einstein, Maxwell and scalar field equations are produced by variation with respect to the metric tensor, the electromagnetic tensor and the scalar field, respectively, and they read

$$G_{\mu\nu} = T_{\mu\nu}, \tag{44}$$

$$\Box\phi + \dot{f}(\phi)F_{\mu\nu}F^{\mu\nu} = 0, \tag{45}$$

$$\partial_\mu\left(\sqrt{-g}\,f(\phi)\,F^{\mu\nu}\right) = 0, \tag{46}$$

where the energy-momentum tensor contains contributions from the scalar and electromagnetic field:

$$\begin{aligned} T_{\mu\nu} = &-\frac{1}{4}g_{\mu\nu}(\nabla\phi)^2 + \frac{1}{2}\nabla_\mu\phi\nabla_\nu\phi \\ &+ f(\phi)\left[\frac{1}{2}g_{\mu\nu}F_{\mu\nu}F^{\mu\nu} - 2g^{\rho\sigma}F_{\mu\rho}F_{\nu\sigma}\right]. \end{aligned} \tag{47}$$

The explicit form of field Equations (44)–(46) for a spherically symmetric line-element can be found in Appendix B.

As in the curvature-induced scenario, for the model to be continuously connected to GR, the property $\dot{f}(\phi_0) = 0$ should be satisfied for some $\phi_0$. Here, we will consider three different forms of the coupling function, which satisfy the aforementioned properties, as indicative cases. They are given by

$$f_e(\phi) = -e^{-\alpha\phi^2}, \tag{48}$$

$$f_q(\phi) = -1 + \alpha\phi^2, \tag{49}$$

$$f_h(\phi) = -\cosh\left(\sqrt{-2\alpha}\phi\right), \tag{50}$$

where the coupling constant $\alpha$ is negative. In this case, by taking perturbations of the scalar equation around a RN background, we find that the requirement for the emergence of a tachyonic instability is equivalent to the condition $\ddot{f}(\phi_0)F^2 > 0$. Here, we consider a purely electric field, namely:

$$A_\mu dx^\mu = V(r)\,dt \Rightarrow F_{\mu\nu}F^{\mu\nu} < 0, \tag{51}$$

which, in turn, requires $\ddot{f}(\phi_0) < 0$. If we also integrate by parts, a second condition is derived, namely $\phi\dot{f}(\phi_0) < 0$.

In order to demonstrate the dependence of the shadow radius on the parameters of the theory, we fix $\alpha$ to different negative values and allow for our code to scan the parameter space for the values of $q \equiv Q_e/M$, where $Q_e$ is the electric charge, for which scalarized solutions exist. The existence line for scalarization is presented in the top left panel of Figure 9. To create this plot, we examine the linear stability of scalar perturbations around the RN background. We decompose the field perturbation as was described in Equation (38), and we follow the same procedure. Following this method, we determine the scalarization thresholds for the first three modes, i.e., for $n = 0$, $n = 1$ and $n = 2$. This yields the minimum value of $|\alpha|$ for a fixed value of $q$ for which we expect spontaneous scalarization to occur. This value appears to be increasing as one increases $n$. It is worth pointing out that since the threshold of scalarization corresponds to small values of $\phi$, it is independent of our choices of the coupling function, accounting for the fact that all of them become identical for small $\phi$.

In the remaining three panels of Figure 9, we present the rescaled black-hole shadow $r_{sh}/M$ in terms of $q$ for the three coupling functions given in Equations (48)–(50). The scalarized solutions depicted refer to the fundamental mode of the scalar field with $n = 0$ and

$\ell = 0$. The black solid line in each of the three plots corresponds to the shadow radius for RN black holes with different parameters $q$. The value of it can be found analytically to be:

$$\frac{r_{\text{sh}}}{M} = \frac{\sqrt{9 - 8q^2} + 3}{\sqrt{2 + \left(\sqrt{9 - 8q^2} - 3\right)/(2q^2)}} \, . \tag{52}$$

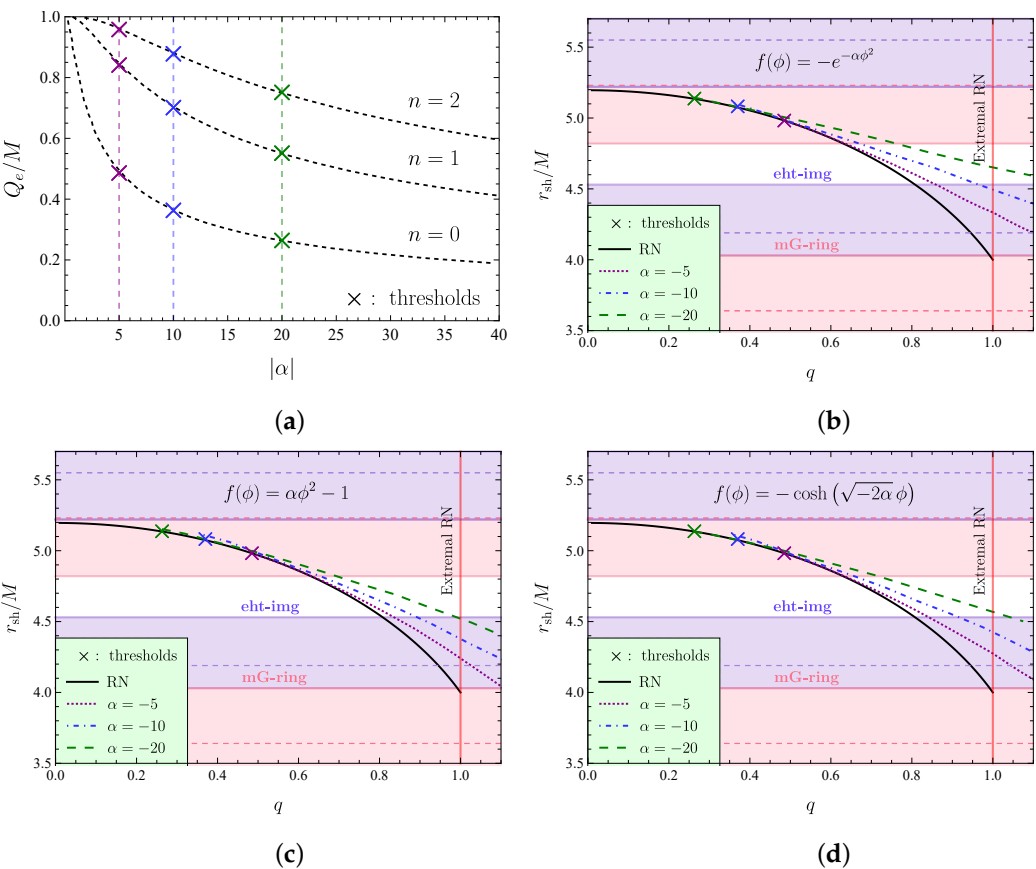

**Figure 9.** (**a**) Onset of scalarization for different overtone numbers. The threshold does not depend on the coupling function. (**b**) Shadow radius for the fundamental mode for spontaneously scalarized EMS black holes with an exponential coupling function $f(\phi) = -e^{-\alpha\phi^2}$, for an s-EM coupling with values $\alpha = \{-5, -10, -20\}$. The solid line corresponds to the GR limit (RN). (**c**) Same as top right but for a quadratic coupling function $f(\phi) = \alpha\phi^2 - 1$. (**d**) Same as top right but for a hyperbolic coupling function of the form $f(\phi) = -\cosh(\sqrt{-2\alpha}\phi)$.

The colored lines correspond to solutions with a different value for the EMS coupling, namely $\alpha = \{-5, -10, -20\}$. The "$\times$" symbol appearing in each colored line corresponds to the scalarization threshold for each $\alpha$. For all three choices of the coupling function, we observe similar results: First, the extremality limit can be exceeded for scalarized solutions, i.e., solutions with $q > 1$ emerge. Second, the charge range appears to increase the more we increase the absolute value of the coupling parameter. This confirms the results appearing in [157–159].

The latter result effectively means that a larger domain in the parameter space of $q$ allows for solutions with a shadow radius lying within the desired bounds. Indeed, as we observe from the three plots in Figure 9 for the three different forms of the coupling function, an increase in $|\alpha|$ decreases the slope of each solution line and thus increases the range of solutions that fall in the white area. These solutions again satisfy all bounds of Table 1 coming from the Sagittarius A* constraints. The more 'conservative' bound, the

*eht-imaging* one, clearly favors solutions with small and up to intermediate values of *q*. On the other hand, the more 'liberal' bound, the *mG-ring* one, tends to favor solutions with intermediate and large values of the charge parameter, including the ones beyond RN extremality. According to these results, charged scalarized solutions can be viable candidates for future-observed black holes. However, on average, they are expected to possess a significant *q* parameter. This does not seem to be the case with Sagittarius A*, for which a very strict upper bound of $q \leq 8.6 \times 10^{-11}$ has been derived [160,161].

## 7. Conclusions

The recent publication of black-hole images by the EHT collaboration gave rise to a novel way to probe the near horizon regime of black holes that is a valuable and complimentary way to test deviations from GR. The data available by the EHT display a bright ring of emission, which surrounds a dark depression that is roughly the size of the black-hole shadow. In order to connect the size of the bright ring to the underlying shadow, one has to use the mass-to-distance ratio, which for the supermassive black hole in the center of our galaxy, SgrA*, is much more accurately known compared to the previously available M87* due to the proximity of SgrA* to the Earth. For this reason, the bounds presented in the recent EHT publication [104] are the strongest to date regarding black-hole metric deviations from GR in the near horizon regime from black-hole imaging.

In this work, we have considered a number of selected theories of modified gravity whose overarching theme is that they predict the existence of black holes bestowed with non-trivial scalar field profiles. In the context of each theory, we have computed the theoretically predicted shadow radii in terms of the fundamental parameters of the theory. These theoretical results may be compared with any existing or future observational bounds in order to probe the validity of the corresponding theories or constrain the parameter space. To this end, here, we have employed, as an indicative example, the observational bound from Sagittarius A* [104] as the most accurate to date.

As there is no clear consensus yet on the spin parameter of Sagittarius A*, we limited our analysis to the spherically symmetric case. For this particular case, the deviation of the black-hole metric from the Schwarzschild scenario is quantified by the fractional deviation *δ* whose bounds were announced in [104] and recreated in the present work in Table 1. Among the various choices displayed in the table, we settled on displaying the results of the image-domain feature extraction procedure *eht-imaging* and the fitting to the analytic model *mG-ring*. Our choices were motivated by the fact that these two constraints represent two very distinct methodologies. In addition, they lie at the two extremes of the spectrum of possible results, with the *eht-imaging* constraints being the most conservative ones allowing only for small deviations from GR and the *mG-ring* constraints being the most liberal ones favoring much larger deviations from GR.

Regarding the theories under consideration, we first focused on the EsGB theory, which is a well-motivated modification of GR that involves higher curvature terms. Our focus in Section 4 was to study generic black holes with non-trivial scalar hair that are regular from the horizon to infinity and for several different choices for the scalar coupling function. We found that, for the linear coupling, the parameter space of the theory cannot be constrained by the EHT observations since the entire range of solutions is either allowed by the *eht-imaging* constraint or excluded by the *mG-ring* constraints within 1-*σ* accuracy. However, for the quadratic and exponential couplings, we found a distinctly different behavior of the solutions with positive and negative coupling parameters. The solutions derived for a positive coupling exhibit the same behavior as in the linear coupling case with the whole set of solutions being allowed by the former EHT constraint and excluded by the latter. On the other hand, solutions with a negative coupling extend over a larger part of the parameter space and may thus be more effectively constrained by the EHT bounds. In these cases, subclasses of solutions that satisfy all EHT constraints within 1-*σ* accuracy could be determined. We also find that special solutions for which the energy momentum tensor component $T_r^r(r)$ can have a local maximum from the horizon to infinity

can only occur for the quadratic coupling in a way that is consistent with the EHT results. In the context of this theory, we also highlighted differences in the shadows between black holes and wormholes. However, a detailed analysis featuring wormhole solutions is left for future work.

Subsequently, in Sections 5 and 6, we turned our attention to spontaneous scalarization. We considered two different scenarios; in the first one, scalarization is associated with the compactness of the object. In this case, we examined, in detail, the effects on the shadow radius from the couplings of a scalar field with curvature invariants (Ricci and GB). We saw that, in principle, the EHT can place significant constraints on the theory depending on the choices of the coupling parameters under examination. For the minimal EsRGB model, we saw that there exists a small region in the parameter space of solutions that satisfies even the tightest combinations of the EHT bounds presented in Table 1. If we also allow for higher-order operator corrections in the EsGB coupling, then the allowed parameter space widens due to the fact that the minimal black-hole mass, in this case, is pushed toward zero.

Finally, in Section 6, we studied scalarization as a result of a non-minimal coupling of a scalar field with the Maxwell tensor. Compared to the RN scenario, we were able to demonstrate that scalarized EMS black holes allow for agreement with the EHT bounds for a broader range of electric charges. Additionally, solutions are retrieved beyond the GR extremality limit with shadow radii within the desired bounds.

Looking to the future, the Next-Generation EHT (ngEHT) project will provide us with significantly sharper images of the shadow of supermassive black holes, such as M87* and SgrA*, and also possibly real-time video of the evolution of the accretion disk around the black-hole horizon. This will usher a whole new era in fundamental physics in the strong-gravity regime while giving birth to a whole new field: imaging and time resolution of black holes on horizon scales [162,163]. It remains to be seen if the preference for a smaller black-hole shadow than the one predicted in the Schwarzschild case will persist in the next generation of experiments and what implications on the viability of modified theories this will entail.

**Author Contributions:** G.A., A.P. and P.K. contributed equally to this paper. All authors have read and agreed to the published version of the manuscript.

**Funding:** This work was supported by IBS under the project code IBS-R018-D1.

**Data Availability Statement:** Data was used from the Event Horizon Telescope collaboration [104].

**Conflicts of Interest:** The authors declare no conflict of interest.

**Appendix A. Equations in EsRGB Theory**

Here we present the equations of motion in the general coupling case of the Einstein-scalar-Ricci–Gauss–Bonnet scenario, as this includes all of the cases considered in Sections 4 (by setting $h(\phi) = 0$) and 5. The spherically symmetric ansatz we chose here has the following form:

$$ds^2 = -A(r)dt^2 + B(r)^{-1}dr^2 + r^2 d\Omega^2 \,. \tag{A1}$$

For this ansatz, the two independent gravitational equations we use plus the scalar equation of motion read:

$$
\begin{aligned}
(t,t): \ & 16B^2\left(\dot{f}\phi'' + \ddot{f}\phi'^2\right) + B'\big[24B\dot{f}\phi' - 4(h+1)r \\
& - 2\phi'\left(4\dot{f} + \dot{h}r^2\right)\big] - B\big[r^2\phi'^2 + 16\dot{f}\phi'' + 16\ddot{f}\phi'^2 \\
& + 4r^2\ddot{h}\phi'^2 + 4\dot{h}r\left(r\phi'' + 2\phi'\right) + 4(h+1)\big] \\
& + 4(h+1) = 0 \,,
\end{aligned}
\tag{A2}
$$

$$(r,r): 24B^2\dot{f}A'\phi' + B\big[-8\dot{f}A'\phi' - 2\dot{h}r^2A'\phi'$$
$$-4(h+1)rA' + Ar\phi'(r\phi' - 8\dot{h}) - 4A(h+1)\big] \tag{A3}$$
$$+4A(h+1) = 0,$$

$$(\phi): 2A^2Br^2\phi'' + 8AB^2\dot{f}A'' - 4A^2\dot{h}(rB'+B-1)$$
$$+\phi'\left(A^2r^2B' + 4A^2Br + ABr^2A'\right) - 4AB\dot{h}rA'$$
$$-2ABA''\left(4\dot{f}+\dot{h}r^2\right) - AA'B'\left(4\dot{f}+\dot{h}r^2\right) \tag{A4}$$
$$+12AB\dot{f}A'B' + BA'^2\left(\dot{h}r^2 - 4(B-1)\dot{f}\right) = 0.$$

### Appendix B. Equations in EMS Theory

For the EMS scalarization model discussed in Section 6, we use the following metric ansatz (in order to be consistent with [157,158]):

$$ds^2 = -N(r)e^{-2\delta(r)}dt^2 + N(r)^{-1}dr^2 + r^2d\Omega^2 \tag{A5}$$

where $N(r) = 1 - 2m(r)/r$, with $m(r)$ being the Misner–Sharp mass [164]. Then, the Einstein (*tt* and *rr*), scalar and electromagnetic Equations (44)–(46) yield:

$$(t,t): m' - \frac{1}{8}r^2\left(1 - \frac{2m}{r}\right)\phi'^2 + \frac{1}{2}e^{2\delta}r^2f\,V'^2 = 0, \tag{A6}$$

$$(r,r): 4\delta' + r\phi'^2 = 0, \tag{A7}$$

$$(\phi): 4r\,(r-2m)\phi'' + r^2(2m-r)\phi'^3 - 8e^{2\delta}r^2V'^2\dot{f}$$
$$+4\left[e^{2\delta}r^3fV'^2 + r\delta'(2m-r) - 2m + 2r\right]\phi' = 0, \tag{A8}$$

$$(em): r^2f\,V' - e^{-\delta}Q_e = 0. \tag{A9}$$

It is then straightforward to solve with respect to $m''$ and $\phi''$, which leaves with a system of ordinary differential equations that can be integrated. The appropriate boundary conditions are found by taking the near-horizon expansions of the functions $m$, $\phi$, $\delta$, $V$.

### Note

[1] Let us note that although we will make use of the bounds on the observed black-hole shadow from Sagittarius A* [104], our analysis will cover also the corresponding bound from the M87* observation [91–98,120,126] as the latter is less stringent and thus easier to satisfy.

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
