# Peer review of "Probing Modified Gravity Theories with Scalar Fields Using Black-Hole Images"

_universe, doi:10.3390/universe9030147_

Round 1
Reviewer 1 Report
The authors discuss modified gravitational theories with a scalar field and investigate their black hole solutions. The shadows are also considered. Constraints on the sizes of black hole shadows are established. Section 3 covers the constraints on EHT observations and shows that the size of the shadow is always smaller than the size of the last stable orbit, and that the deviation parameter is always negative. The authors provide interesting arguments why it is not necessary to consider rotating black hole metrics at the current moment. Section 4 considered scalar-tensor gravity with the Gauss-Bonnet term, focussing on the solutions with different coupling functions. The section also imposed constraints on parameters based on two different methods. However, the solution of the wormhole type does not satisfy both constraints. In section 5, the authors conclude that the EGB model with a quadratic coupling can lead to stable black hole metrics with spontaneous scalarization. Using constraints on the mass and radius of black hole shadows, the authors conclude that Sgr A* may be a potential candidate for a black hole with spontaneous scalarization. Additional experimental data and more accurate theoretical constraints are required for a final conclusion. Section 6 describes the modelling of scalar fields and electromagnetic fields around black holes in the framework of the Einstein-Maxwell-scalar (EM) model. However, the results of this section cannot be directly applied to Sgr A* as this object is a charged rotating Kerr black hole. Overall, the article is interesting and relevant in the field of research into extended theories of gravity and black holes. The authors conducted a serious investigation, described various modified gravity theories, and compared their predictions with observational data for Sgr A*. In addition, the article proposed methods to constrain extended gravity models using various approaches.
However, it should be noted that the authors did not consider rotating black holes and did not compare the results obtained with observational data for Sgr A* taking rotation into account. To my mind this is a disadvantage of the work, as real black holes rotate. The consideration of the rotation is necessary to compare with observational data but authors from the beginning gave their arguments against this.
The only critical note is that the choice of the coupling function used in the paper requires additional arguments.
As a result the paper can be accepted for publication after adding the notes on coupling functions choice.
Reviewer 2 Report
Please, find the attached file.

Reviewer 3 Report
The authors considered three classes of the scalar-tensor theory of gravity to analyze the properties of shadows of black holes and wormholes. In the weak-field limit, these modified gravitational theories have predictions similar to GR's. To compare these theories, one should employ data from a strong-field regime.
In this work, spherically symmetric spacetime is considered and the fractional deviation $\delta$ is used to measure the deviation of the black hole solutions from the Schwarzschild solution.
I think this paper is well written and the results are correct. It can be accepted for publication in the journal Universe.
Author Response
We would like to thank the Reviewer 3 for the careful reading of our manuscript and the positive assessment of our work. For his/her information, we note that, in the revised version of our manuscript, we have only added some clarifications and explanatory comments as well as a few references while our analysis and results have remained unchanged.